# Association between HTLV-1/2 infection and COVID-19 severity in a migrant Shipibo-Konibo population in Lima, Peru

**Fátima Avila Dextre**[1], **Bryan Morales Álvarez**[1], **Paulo Aguirre Castañeda**[1], **Isaac Efrain Alva**[2], **Giovanni López**[3], **Alvaro Schwalb**[3,4], **Eduardo Gotuzzo**[1,3]*

1 School of Medicine, Universidad Peruana Cayetano Heredia, Lima, Peru, 2 School of Public Health, Universidad Peruana Cayetano Heredia, Lima, Peru, 3 Instituto de Medicina Tropical Alexander von Humboldt, Universidad Peruana Cayetano Heredia, Lima, Peru, 4 Department of Infectious Disease Epidemiology, London School of Hygiene & Tropical Medicine, London, United Kingdom

* eduardo.gotuzzo@upch.pe

**Data Availability Statement:** All relevant data are within the manuscript and its Supporting information files.

## Abstract

### Objectives

The migrant community of the Shipibo-Konibo indigenous people in Lima, Peru were extremely vulnerable during the COVID-19 pandemic. Additionally, infection with human T-cell lymphotropic virus type 1 and 2 (HTLV-1/2) is endemic in this population causing immunosuppression. The aim of the study was to describe the association between HTLV-1/2 infection and the clinical severity of COVID-19.

### Methods

This was a cross-sectional descriptive study involving a survey of adult Shipibo-Konibo indigenous migrants residing in Cantagallo-Rímac who were identified as suspected or confirmed cases of COVID-19. Blood samples were collected for SARS-CoV-2 antibody and HTLV-1/2 ELISA testing. A confirmatory Western Blot test was performed for those with a positive ELISA test.

### Results

A total of 182 individuals were surveyed and sampled. No significant association was found between HTLV-1/2 infection and the clinical severity of COVID-19. The prevalence of HTLV-1/2 was 8.8% (95%CI: 5.0–14.1) with Western Blot. Age was the only statistically significant risk factor for developing a more severe form of COVID-19 (OR: 1.03; 95%CI: 1.00–1.06; p = 0.032).

### Conclusions

There was no association found between HTLV-1/2 infection and the clinical severity of COVID-19. The prevalence of HTLV-1/2 infection in the Shipibo-Konibo population is high and warrants continuous monitoring in the advent of other infectious disease outbreaks and the development of HTLV-associated comorbidities.

**Funding:** The authors received no specific funding for this work.

**Competing interests:** The authors have declared that no competing interests exist.

# Background

The Shipibo-Konibo are one of the largest indigenous communities in the Peruvian Amazon, with a population of over 34,000 individuals [1]. This group has traditionally resided along the Ucayali River in the Amazon rainforest in Peru until a huge exodus towards the cities of Pucallpa and Lima began in 1990 [2]. Since then, approximately 300 families have been established in the shantytown of Cantagallo, in the Lima Metropolitan Area [2]. This settlement, with precarious homes and with poor sanitation services, was highly affected by the COVID-19 pandemic [3]. In May 2020, in a mass campaign by the Ministry of Health, 72.5% of Shipibo-Konibo migrants in Cantagallo tested positive for SARS-CoV-2. [4]. Later, the neighbourhood was considered the highest COVID-19 hotspot in the country and the government imposed a military-guarded epidemiological fencing in the area further limiting access to basic needs and healthcare services [3]. Consequently, risk factors for severe COVID-19 were more prominent in this indigenous people compared to the general population.

Apart from the existing social vulnerabilities within this population, a high prevalence of human T-lymphotropic virus (HTLV) 1/2 has been observed [5, 6]. HTLV-1/2 is a greatly oncogenic retrovirus that is often overlooked despite its associated elevated morbidity and mortality [7]. While over 90% of those infected remain asymptomatic, HTLV-1/2 infection can cause an important degree of immunosuppression leading to strongyloidiasis, crusted scabies, and tuberculosis [8–10]. Thereby, hypotheses were postulated regarding the potential influence of HTLV infection on the severity of COVID-19 [11].

The high prevalence of HTLV-1/2 and the number of cases of COVID-19 reported among the Shipibo-Konibo migrants suggest a possible association between these two viruses. Some studies have already sought to report on the outcomes of the co-infection [12, 13]; however, there is a further need to evaluate this within this neglected population. Thus, we aimed to evaluate risk factors for COVID-19 severity in the Shipibo-Konibo community focusing on the role HTLV-1/2 infection plays in the development of severe COVID-19.

# Methods

## Study design and setting

We conducted a cross-sectional study in the migrant Shipibo-Konibo community in Cantagallo, district of Rimac, Lima Metropolitan Area between July 2021 and April 2022.

## Study population

We recruited adult (>18 years) individuals of the Shipibo-Konibo indigenous people living in Cantagallo. Since at one point the neighbourhood was considered the highest COVID-19 hotspot in the country, all inhabitants met the criteria of a probable or confirmed case of COVID-19, following the definitions of the National Centre for Epidemiology, Prevention and Disease Control (CDC MINSA) [14]. A probable case was defined based on clinical or epidemiological criteria, and a confirmed case was defined based on molecular or rapid testing for SARS-CoV-2 (see Table 1 for complete definitions) [14]. Severity of COVID-19 is divided into mild, moderate, and severe (see Table 2 for severity criteria) [14]. We defined overcrowding for participants as a condition where the number of rooms divided by number of inhabitants in the household was greater than or equal to 3. We excluded individuals participating in clinical trials evaluating the efficacy and safety of COVID-19 vaccine candidates.

**Table 1. COVID-19 case definitions.**

| Probable case | Confirmed case |
|---|---|
| Individual with an acute respiratory infection who presents with two or more of the following symptoms: cough, sore throat, shortness of breath, nasal congestion, and fever.<br>AND<br>Contact with a confirmed case of COVID-19 infection or travel history outside the country or to cities in Peru with community transmission of COVID-19 within 14 days prior to the onset of symptoms, or individuals with severe acute respiratory infection (fever, cough, and respiratory distress) requiring hospitalisation. | Probable case with a positive laboratory test for COVID-19, either RT-PCR from respiratory samples or a rapid antibody test (IgM/IgG positive).<br>OR<br>Asymptomatic contact with a positive laboratory test for COVID-19. |

Criteria of the National Centre for Epidemiology, Prevention and Disease Control (CDC MINSA) [14].

## Study procedures

Participants were enrolled by convenience sampling due to the geographic accessibility and availability of the residents in Cantagallo. A household visit was conducted to obtain informed consent and administer the questionnaire designed for the study (S1 Questionnaire). The questionnaire covered information on sociodemographic characteristics, clinical history, and items on HTLV-1/2 and COVID-19. At the end of the visit, blood samplings were scheduled in the communal area of Cantagallo at a later date.

## Laboratory procedures

Blood was analysed for total antibodies against SARS-CoV-2 (ECLIA, Electro-Chemiluminescence Immunoassay), haematocrit and glucose. An enzyme-linked immunosorbent assay (ELISA) for HTLV-1/2 (Architet HTLV-I+II, Abbott, Germany) was processed in the "Tropicales Lab" at the Instituto de Medicina Tropical Alexander von Humboldt. All serological samples that were positive for HTLV-1/2 underwent a confirmatory Western Blot (WB) test at the Viral Immunology Section, Neuroimmunology and Neurovirology Division (NND) of the National Institute of Health (NIH). The results were shared with the participants in-person and in writing; individuals who tested positive were provided with instructions for further monitoring.

**Table 2. COVID-19 severity.**

| Any individual with acute respiratory illness | | |
|---|---|---|
| **Mild** | **Moderate** | **Severe** |
| Presenting at least two of the following signs and symptoms:<br>• Cough<br>• General discomfort<br>• Sore throat<br>• Fever<br>• Nasal congestion | Presenting any of the following criteria:<br>• Dyspnea or difficulty breathing<br>• Respiratory rate higher than 22 breaths per minute<br>• Altered level of consciousness<br>• Arterial hypotension or shock<br>• Clinical and/or radiological signs of pneumonia<br>• Lymphocyte count less than 1000 cell/μL | Presenting two or more of the following criteria:<br>• Respiratory rate > 22 breaths per minute or $PaCO_2 < 32$ mmHg<br>• Altered level of consciousness<br>• Systolic blood pressure < 100 mmHg or mean arterial pressure (MAP) <65 mmHg<br>• $PaO_2 < 60$ mmHg or PaFi < 300<br>• Clinical signs of muscle fatigue, nasal flaring, use of accessory muscles, thoraco-abdominal imbalance<br>• Serum lactate > 2 mmol/L |

Criteria of the National Centre for Epidemiology, Prevention and Disease Control (CDC MINSA) [14].

## Statistical analysis

Anonymised data was analysed using Stata SE v17.0 (StataCorp., US). Frequencies and percentages were reported for categorical variables and medians with the interquartile range (IQR) for continuous variables. The chi-squared and Fisher's exact test were applied for comparisons of proportions between two discrete variables. The Mann-Whitney U test was used to compare differences in continuous variables. We used logistic regression to evaluate the association between HTLV-1/2 infection with severe COVID-19, reporting the odds ratio (OR) with their corresponding 95% confidence intervals (CI). The model assumed a binomial distribution of the response variable and employed the logit link function. Details on the model formula are provided in S1 File. Variable selection was guided by clinical and epidemiological relevance, supported by theoretical justifications and prior research findings; no statistical method or algorithm were used to select the logistic regression covariates. The performance of the model was assessed using the area under the ROC curve, which was 0.68, demonstrating the moderate discriminative ability of the model to correctly classify patient outcomes.

## Ethical considerations

Study protocol was approved by the Research Ethics Committee of Universidad Peruana Cayetano Heredia (CIE-UPCH) under reference number 204694. The study was also approved by the Northern Lima Directorate of Integrated Health Networks (DIRIS, acronym in Spanish) of the Ministry of Health (Ref. N˚-971-2021-MINSA/DIRIS.LN/1). Additionally, meetings were held with the leaders of the Association of Shipibo Artisans Residents in Lima (ASHIREL), the Shipibo-Konibo Urban Community Association of Lima Metropolitan Area (ACUSHI-KOLM), the Shipibo Housing Association in Lima (AVSHIL) and the Shipibo-Konibo Community Association of Cantagallo (ACC) where the corresponding approvals for the development of the study were obtained. Written or thumbprint informed consent was obtained from all participants.

## Inclusivity in global research

Additional information regarding the ethical, cultural, and scientific considerations specific to inclusivity in global research is included in S1 Checklist.

# Results

During the study, 217 participants from the migrant Shipibo-Konibo community in Cantagallo were surveyed. Most of the sampled participants were women (68.1%), and the median age was 34 years (IQR: 25–45). Blood sampling for SARS-CoV-2 and HTLV-1/2 were performed for 182 participants (83.9%). The ELISA results for HTLV-1/2 showed 38 positives. The sera from participants with a positive ELISA test for HTLV (n = 38) were sent to the NIH for confirmation through WB, with 27 samples testing positive. Out of these 27, 12 participants already had a prior HTLV-1/2 diagnosis before the start of the study, thus resulting in a prevalence of 8.8% (95%CI: 5.0–14.1). The results showed HTLV-1 in eight samples (4.7%; 95%CI: 2.1–9.1), HTLV-2 in seven samples (4.1%; 95%CI: 1.7–8.3), four were seroindeterminate, and seven were seronegative. Among those WB HTLV-1/2 positive, the median age was 46 years (IQR: 35–53) (Table 3). None of the participants with WB HTLV-1/2 positive results presented any comorbidities associated with HTLV-1/2 infection, such as tropical spastic paraparesis and adult T-cell leukaemia/lymphoma.

The ECLIA for antibodies against SARS-CoV-2 were positive in 145 (79.7%) and negative in 37 (20.3%) participants. More than two-thirds of the participants (70.9%) reported having

**Table 3. Characteristics of the Shipibo-Konibo migrant community in Cantagallo, Lima.**

| Characteristics | All participants (n = 182) | WB HTLV positive (n = 27) | HTLV negative and indeterminate[†] (n = 155) | P value |
|---|---|---|---|---|
| Age, years, median (IQR) | 34 (25–45) | 46 (35–53) | 32 (25–42) | 0.003 |
| Male sex | 58 (31.9) | 8 (29.6) | 50 (32.3) | 0.787 |
| Overcrowding[‡] | 80 (44.0) | 15 (55.6) | 65 (42.0) | 0.188 |
| Comorbidities | 53 (29.1) | 11 (40.7) | 42 (27.1) | 0.150 |
| Diabetes mellitus | 14 (7.7) | 4 (14.8) | 10 (6.5) | 0.134* |
| Anaemia | 9 (5.0) | 2 (7.4) | 7 (4.5) | 0.523 |
| Hypertension | 5 (2.8) | 1 (3.7) | 4 (2.6) | 0.556* |
| Asthma | 5 (2.8) | 1 (3.7) | 4 (2.6) | 0.556* |
| Dyslipidaemia | 2 (1.1) | 0 (0.0) | 2 (1.3) | - |
| Others | 30 (16.5) | 6 (22.2) | 24 (15.5) | 0.555 |

Values are n (%) unless otherwise indicated. P value was calculated using chi-squared test unless specified otherwise.

[†]HTLV negative includes participants with negative and indeterminate WB results and those who did not take the WB test.

[‡]Number of rooms/number of inhabitants ≥3.

*Fisher's exact test was used. HTLV-1/2: human T-lymphotropic virus type 1 and 2; WB: Western Blot; IQR: Interquartile range; COVID-19: Coronavirus Disease 2019

experienced COVID-19 symptoms since March 2020 (the month when the first case of COVID-19 was confirmed in Peru), with no difference according to the presence of HTLV-1/2 infection. The most frequent symptoms in those with WB HTLV-1/2 positivity were headache (90.5%), myalgia (80.9%), and dysgeusia (76.2%). Regarding the clinical severity of COVID-19 in the WB HTLV-positive group, it was observed that approximately half had a mild form (51.9%). No participants developed a severe form of COVID-19 (Table 4). Surprisingly, among the vaccinated group, there was a higher proportion of individuals that had moderate COVID-19 severity compared to the unvaccinated group.

COVID-19 vaccination began during the study period (August-September 2021), and out of the 73 participants who claimed to have been vaccinated, 64 (87.7%) had positive antibodies against SARS-CoV-2. Concerning the clinical severity of COVID-19 among those with a positive antibody test, it was observed that 38 (26.2%) were asymptomatic, 58 (40.0%) had mild symptoms, and 49 (33.8%) had moderate symptoms (Table 5).

Furthermore, risk factors associated with the clinical severity of COVID-19 were studied. In the multivariate analysis, age was statistically significant, with an adjusted OR of 1.03 (95% CI: 1.00–1.06; p = 0.032) for developing moderate COVID-19 for each unit increase in age. Finally, being infected with HTLV-1/2 (confirmed by WB) was not statistically significant (adjusted OR: 0.55; 95%CI: 0.20–1.50; p = 0.237) (Table 6). As a sensitivity analysis, we evaluated risk factors considering individuals without comorbidities. Similar results were found with age being the only statistically significant risk factor associated with COVID-19 severity (Table 7). Additionally, another sensitivity analysis was conducted where the impact of SARS-CoV-2 vaccination on the severity of COVID-19 was evaluated showing a significant but inverse association than expected (aOR: 2.30; 95%CI: 1.14–4.65; p = 0.020) (S1 File).

## Discussion

Our study did not find an association between HTLV-1/2 infection and the risk of developing a more severe form of COVID-19. In particular, we identified 27 individuals with HTLV-1/2 infection, of which only seven were classified as moderate COVID-19 cases due to the presence of dyspnoea. Despite the abundance of literature since the first COVID-19 case was reported,

**Table 4. Information on COVID-19 in residents of the Shipibo-Konibo migrant community in Cantagallo, Lima.**

| Characteristics | All participants (n = 182) | WB HTLV positive (n = 27) | WB HTLV negative and indeterminate[†] (n = 155) | P value | Vaccinated (n = 73) | Unvaccinated (n = 109) | P value |
|---|---|---|---|---|---|---|---|
| **Total antibodies test against SARS-CoV-2[a]** | | | | | | | |
| Positive | 145 (79.7) | 22 (81.5) | 123 (79.4) | 0.800 | 64 (87.7) | 81 (74.3) | 0.028 |
| Negative | 37 (20.3) | 5 (18.5) | 32 (20.7) | 0.800 | 9 (12.3) | 28 (25.7) | 0.028 |
| **Presence of COVID-19 symptoms** | 129 (70.9) | 21 (77.8) | 108 (69.7) | 0.393 | 55 (75.3) | 74 (67.9) | 0.278 |
| Cough | 70 (54.3) | 6 (28.6) | 64 (59.3) | 0.010 | 34 (61.8) | 36 (48.7) | 0.138 |
| Fever | 94 (72.9) | 15 (71.4) | 79 (73.2) | 0.871 | 41 (74.6) | 53 (71.6) | 0.712 |
| Dyspnoea/shortness of breath | 55 (42.6) | 7 (33.3) | 48 (44.4) | 0.346 | 30 (54.6) | 25 (33.8) | 0.018 |
| Headache | 103 (79.8) | 19 (90.5) | 84 (77.8) | 0.184 | 45 (81.8) | 58 (78.4) | 0.630 |
| Diarrhoea | 31 (24.0) | 4 (19.1) | 27 (25.0) | 0.781* | 15 (27.3) | 16 (21.6) | 0.534 |
| Anosmia | 98 (75.9) | 15 (71.4) | 83 (76.9) | 0.595 | 45 (81.8) | 53 (71.6) | 0.180 |
| Dysgeusia | 101 (78.3) | 16 (76.2) | 85 (78.7) | 0.798 | 48 (87.3) | 53 (71.6) | 0.033 |
| Fatigue | 70 (54.3) | 9 (42.9) | 61 (56.5) | 0.251 | 31 (56.4) | 39 (52.7) | 0.680 |
| Rhinorrhoea | 67 (51.9) | 10 (47.6) | 57 (52.8) | 0.665 | 33 (60.0) | 34 (45.9) | 0.114 |
| Nauseas/vomiting | 35 (27.1) | 9 (42.9) | 26 (24.1) | 0.077 | 17 (30.9) | 18 (24.3) | 0.406 |
| Myalgias | 100 (77.5) | 17 (80.9) | 83 (76.9) | 0.680 | 45 (81.8) | 55 (74.3) | 0.313 |
| **COVID-19 severity[‡]** | | | | | | | |
| Asymptomatic | 53 (29.1) | 6 (22.2) | 47 (30.3) | 0.393 | 18 (24.7) | 35 (32.1) | 0.278 |
| Mild | 74 (40.7) | 14 (51.9) | 60 (38.7) | 0.199 | 25 (34.3) | 49 (44.9) | 0.149 |
| Moderate | 55 (30.2) | 7 (25.9) | 48 (30.9) | 0.599 | 30 (41.1) | 25 (22.9) | 0.009 |

Values are n (%) unless otherwise indicated. P value was calculated using chi-squared test unless specified otherwise.

[†]HTLV negative includes participants with negative and indeterminate WB results and those who did not take the WB test.

[‡]Severity classification according to the Technical Document: Criteria of the National Centre for Epidemiology, Prevention and Disease Control (CDC MINSA) [14].

*Fisher's exact test was used. HTLV-1/2: human T-lymphotropic virus type 1 and 2; WB: Western Blot; IQR: Interquartile range; COVID-19: Coronavirus Disease 2019

there are no conclusive studies on this association [15]. The immunology of the co-infection suggests an increased risk of COVID-19 disease severity due to dysregulated T-cell functioning and increased cytokine release in patients with HTLV-1/2 infection [11]. Isolated cases of the co-infection have been described in Brazil and Japan [12, 16, 17]. A case report from Japan described a severe case of COVID-19 in a 73-year-old patient who had not been previously

**Table 5. Information on the presence of antibodies against COVID-19 in residents of the Shipibo-Konibo migrant community in Cantagallo, Lima.**

| | Positive SARS-CoV-2 antibodies (n = 145) | Negative SARS-CoV-2 antibodies (n = 37) | P value |
|---|---|---|---|
| **Vaccination status** | | | |
| Unvaccinated | 81 (74.3) | 28 (25.7) | 0.028 |
| Vaccinated | 64 (87.7) | 9 (12.3) | 0.028 |
| **COVID-19 severity[‡]** | | | |
| Asymptomatic | 38 (26.2) | 15 (40.5) | 0.087 |
| Mild | 58 (40.0) | 16 (43.2) | 0.720 |
| Moderate | 49 (33.8) | 6 (16.2) | 0.038 |

Values are n (%). P value was calculated using chi-squared test.

[‡]Severity classification according to the Technical Document: Criteria of the National Centre for Epidemiology, Prevention and Disease Control (CDC MINSA) [14].

**Table 6. Risk factors associated with COVID-19 severity in residents of the Shipibo-Konibo migrant community in Cantagallo, Lima.**

| Variables | Asymptomatic and mild COVID-19 (n = 127) | Moderate COVID-19 (n = 55) | Odds ratio (95%CI) | Adjusted odds ratio (95%CI) | P value |
|---|---|---|---|---|---|
| Age, years, median (IQR) | 31 (24–43) | 38 (32–47) | 1.03 (1.00–1.06) | 1.03 (1.00–1.06) | 0.032 |
| Male gender | 40 (31.5) | 18 (32.7) | 1.06 (0.53–2.08) | 1.14 (0.55–2.35) | 0.731 |
| Overcrowding | 53 (41.7) | 27 (49.1) | 1.35 (0.71–2.54) | 1.41 (0.72–2.78) | 0.319 |
| HTLV-1/2[†] | 20 (15.7) | 7 (12.7) | 0.78 (0.30–1.97) | 0.55 (0.20–1.50) | 0.237 |
| Diabetes mellitus | 8 (6.3) | 6 (10.9) | 1.82 (0.60–5.52) | 1.62 (0.43–6.07) | 0.470 |
| Asthma | 4 (3.1) | 1 (1.8) | 0.57 (0.06–5.21) | 0.18 (0.01–3.14) | 0.237 |
| Anaemia[‡] | 18 (14.3) | 10 (18.5) | 1.36 (0.58–3.18) | 1.49 (0.61–3.6) | 0.378 |
| Arterial hypertension | 1 (0.8) | 4 (7.3) | 9.88 (1.08–90.57) | 13.13 (0.94->99.99) | 0.060 |
| Dyslipidaemia | 1 (0.8) | 1 (1.8) | 2.33 (0.14–38.0) | 0.47 (0.01–31.0) | 0.722 |

Values are n (%) unless otherwise indicated.

[†]According to HTLV-1/2 WB and ELISA result.

[‡]According to haematocrit values obtained in this study (normal values: males = 41–53%, females = 36–46%). IQR: Interquartile range.

diagnosed with HTLV-1; the patient died 3-months after symptom onset due to uncontrollable infections while on continuous ventilatory support [12]. On the other hand, another case has been reported in a patient with adult T-cell leukaemia who, despite presenting immunosuppression, did not develop severe pneumonia due to COVID-19 [17]. Furthermore, two patients with known HTLV-1/2 infection in Brazil also presented with mild COVID-19 severity [16]

Other studies have also investigated the coinfection of SARS-CoV-2 and other viruses, finding that this is associated with higher mortality compared to coinfection with bacteria [18]. Particularly, a systematic review demonstrated that HIV-infected patients have a greater risk of SARS-CoV-2 infection and higher mortality from COVID-19 [19]. While both HIV and HTLV-1/2 are closely related retroviruses, the latter has not been prominently featured in the literature since the emergence of COVID-19.

The prevalence of HTLV-1/2 infection was 8.8% in this selected population of Shipibo-Konibo migrants in Cantagallo with a prior diagnosis of COVID-19. However, it is important to note that this sample may not be representative of the entire population as it was done through convenience sampling. In the general Peruvian population, the prevalence of HTLV-1/2 infection has been estimated to be 2.9%; nonetheless, there are varying prevalence rates between regions and populations surveyed [20]. In 2013, Blas et al. reported a prevalence of

**Table 7. Risk factors associated with COVID-19 severity in residents without comorbidities of the Shipibo-Konibo migrant community in Cantagallo, Lima.**

| Variables | Asymptomatic and mild COVID-19 (n = 95) | Moderate COVID-19 (n = 34) | Odds ratio (95%CI) | Adjusted odds ratio (95%CI) | P value |
|---|---|---|---|---|---|
| Age, years, median (IQR) | 31 (22–35) | 37 (32–47) | 1.06 (1.02–1.09) | 1.06 (1.03–1.11) | 0.001 |
| Male gender | 30 (31.6) | 13 (38.2) | 1.34 (0.58–3.02) | 1.76 (0.72–4.30) | 0.209 |
| Overcrowding | 42 (44.2) | 19 (55.9) | 1.59 (0.73–3.56) | 1.60 (0.69–3.78) | 0.272 |
| HTLV-1/2[†] | 11 (11.6) | 5 (14.7) | 1.32 (0.39–3.96) | 0.89 (0.24–2.95) | 0.861 |

Values are n (%) unless otherwise indicated.

[†]According to HTLV-1/2 WB and ELISA result.

[‡]According to haematocrit values obtained in this study (normal values: males = 41–53%, females = 36–46%). IQR: Interquartile range.

9.7% among women of the Shipibo-Konibo indigenous people in the Lima and Ucayali regions [5]. The high prevalence and coexistence of HTLV-1 and HTLV-2 in this particular population is also noteworthy [21].

Seroindeterminate results by WB are defined as an incomplete band pattern to the Gag or Env proteins of HTLV [22]. According to Yao et al., this result may reflect prior exposure to HTLV-1 associated with a partial immune response, meaning an 'immune memory' of exposure [22]. The presence of 18.4% false positives (ELISA + and WB -) and 10.5% seroindeterminate cases (ELISA + and WB indeterminate) in our sample underscores the importance of using a confirmatory test to avoid diagnostic errors and eliminate seroindeterminate blood products from circulation [23].

Clinical manifestations of COVID-19 are many; although fever, cough, and fatigue are most common [24]. A series of cases showed no difference in the presence of COVID-19 symptoms between patients with HTLV-1/2 infection and the general population [25]. This differs from our findings where the most frequent symptoms included headache, myalgia, dysgeusia, and anosmia. A meta-analysis suggested that the presence of dyspnoea is associated with an unfavourable progression of the disease [26]. This aligns with our study findings, as dyspnoea was observed in less than 50% of our population, and no severe cases were identified. Age was the only variable with statistical significance in developing moderate COVID-19 (OR: 1.03; 95%CI: 1.00–1.06; p = 0.002), similar to the findings in a retrospective cohort study, where individuals aged 60 or over were associated with worse disease outcomes [27].

During our study, the COVID-19 vaccination campaign had begun for the 40-year-old age group in August-September 2021. Even though indigenous communities were considered a vulnerable population and were eligible to be offered the vaccine earlier, this was not the case for the urban migrants in Cantagallo, who were vaccinated at the same time as the general population, according to their age group [28]. This is evident in our study, as only 73 (40.1%) of participants claimed to have been vaccinated. Furthermore, 64 (44.1%) out of the 145 participants who tested positive for COVID-19 antibodies had been vaccinated, which adds uncertainty to whether the test was positive due to SARS-CoV-2 infection, false positivity, or vaccination [29].

This study had several limitations. Firstly, it is likely that we were unable to identify individuals who previously had severe COVID-19 as some may have had lasting complications and some may have died. It is worth noting that there had already been three reported deaths in Cantagallo when the epidemiological fencing took place, and likely leading to more fatalities in the first months of the pandemic [3]. The disparity in age distribution between the HTLV-1/2 positive and negative groups could pose a limitation because the more severe COVID-19 outcomes observed in the HTLV-1/2 positive group might be attributed to their older age rather than not their HTLV-1/2 status. Furthermore, selection bias was introduced due to convenience sampling (e.g., most of our sample were women, as men were encountered less frequently in the community due to work) and due to some participants being sought out for this study as they were known individuals with HTLV-1/2. As a result, 12 (44.4%) out of the 27 WB HTLV-1/2 positive individuals were not new diagnoses, nor randomly selected. Finally, recall bias might have been present as the information collected was based on the participants' memories on their symptoms since March 2020.

## Conclusions

Despite HTLV-1/2 infection not being associated with COVID-19 severity in our study, like other immunosuppressive conditions, it is likely to be a risk factor for developing a more severe course. The importance of larger studies to reach that conclusion is highlighted. The

experience of the migrant Shipibo-Konibo indigenous people in Cantagallo during the COVID-19 pandemic should emphasise the need for efficient governmental aid in arduous times. Furthermore, the high prevalence of HTLV-1/2 infection warrants continuous monitoring in the advent of other infectious disease outbreaks and development of HTLV-associated comorbidities.

## Supporting information

**S1 File. Supplementary material.**
(DOCX)

**S1 Data. Database and dictionary.**
(XLSX)

**S1 Questionnaire. Study questionnaire.**
(PDF)

**S1 Checklist. Questionnaire on inclusivity in global research.**
(DOCX)

## Acknowledgments

The authors would like to acknowledge the support of Diana Ancón for the translation into the Shipibo-Konibo language to reinforce the key messages, Aydee Trebejo for the field work with the participants and Dr Steven Jacobson for contributing to the development of the study.

## Author Contributions

**Conceptualization:** Fátima Avila Dextre, Bryan Morales Álvarez, Paulo Aguirre Castañeda, Isaac Efrain Alva, Alvaro Schwalb, Eduardo Gotuzzo.

**Data curation:** Fátima Avila Dextre, Bryan Morales Álvarez, Paulo Aguirre Castañeda, Alvaro Schwalb, Eduardo Gotuzzo.

**Formal analysis:** Fátima Avila Dextre, Bryan Morales Álvarez, Paulo Aguirre Castañeda, Alvaro Schwalb, Eduardo Gotuzzo.

**Funding acquisition:** Fátima Avila Dextre, Bryan Morales Álvarez, Paulo Aguirre Castañeda, Eduardo Gotuzzo.

**Investigation:** Fátima Avila Dextre, Bryan Morales Álvarez, Paulo Aguirre Castañeda, Giovanni López, Alvaro Schwalb, Eduardo Gotuzzo.

**Methodology:** Fátima Avila Dextre, Bryan Morales Álvarez, Paulo Aguirre Castañeda, Alvaro Schwalb, Eduardo Gotuzzo.

**Project administration:** Fátima Avila Dextre, Bryan Morales Álvarez, Paulo Aguirre Castañeda, Isaac Efrain Alva, Alvaro Schwalb, Eduardo Gotuzzo.

**Resources:** Fátima Avila Dextre, Bryan Morales Álvarez, Paulo Aguirre Castañeda, Isaac Efrain Alva, Eduardo Gotuzzo.

**Software:** Fátima Avila Dextre, Bryan Morales Álvarez, Paulo Aguirre Castañeda, Alvaro Schwalb, Eduardo Gotuzzo.

**Supervision:** Fátima Avila Dextre, Bryan Morales Álvarez, Paulo Aguirre Castañeda, Alvaro Schwalb, Eduardo Gotuzzo.

**Validation:** Fátima Avila Dextre, Bryan Morales Álvarez, Paulo Aguirre Castañeda, Alvaro Schwalb, Eduardo Gotuzzo.

**Visualization:** Fátima Avila Dextre, Bryan Morales Álvarez, Paulo Aguirre Castañeda, Alvaro Schwalb, Eduardo Gotuzzo.

**Writing – original draft:** Fátima Avila Dextre, Bryan Morales Álvarez, Paulo Aguirre Castañeda, Alvaro Schwalb, Eduardo Gotuzzo.

**Writing – review & editing:** Fátima Avila Dextre, Bryan Morales Álvarez, Paulo Aguirre Castañeda, Alvaro Schwalb, Eduardo Gotuzzo.

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
