## [Decision Letter · Decision Letter 0]

27 Dec 2023

PGPH-D-23-02074

Association between HTLV-1/2 infection and COVID-19 severity in a migrant Shipibo-Konibo population in Lima, Peru

Dear Dr. Avila,

Thank you for submitting your manuscript to PLOS Global Public Health. Firstly, we would like to apologize for the delay in processing your manuscript. It has been exceptionally difficult to secure reviewers to evaluate your study. We have now received one completed review, which is available below. The reviewer has raised significant scientific concerns about the study that need to be addressed in a revision.

Please note that we have only been able to secure a single reviewer to assess your manuscript. We are issuing a decision on your manuscript at this point to prevent further delays in the evaluation of your manuscript. Please be aware that the editor who handles your revised manuscript might find it necessary to invite additional reviewers to assess this work once the revised manuscript is submitted. However, we will aim to proceed on the basis of this single review if possible. 

We look forward to receiving your revised manuscript.

Kind regards,

Miquel Vall-llosera Camps

Staff Editor

Journal Requirements:

2. We ask that a manuscript source file is provided at Revision. Please upload your manuscript file as a .doc, .docx, .rtf or .tex.

4. Please amend your Data Availability Statement and indicate where the data may be found.

Reviewers' comments:

Reviewer's Responses to Questions

**Comments to the Author**

1. Does this manuscript meet PLOS Global Public Health’s publication criteria? Is the manuscript technically sound, and do the data support the conclusions? The manuscript must describe methodologically and ethically rigorous research with conclusions that are appropriately drawn based on the data presented.

Reviewer #1: Yes

2. Has the statistical analysis been performed appropriately and rigorously?

Reviewer #1: Yes

3. Have the authors made all data underlying the findings in their manuscript fully available (please refer to the Data Availability Statement at the start of the manuscript PDF file)?

Reviewer #1: Yes

4. Is the manuscript presented in an intelligible fashion and written in standard English?

Reviewer #1: Yes

5. Review Comments to the Author

Reviewer #1: Dear author,

The manuscript is well written and the topic is interesting but there are some methodological issues. First, please explain the criteria for identifying mild, moderate and severe cases. Second, please perform a separate analysis for patients without comorbidities as the mentioned comorbidities may had affected the severity of disease. Third, please explain whether the HTLV positive patients had ATL or HAM/TSP.

Best Regards,

6. PLOS authors have the option to publish the peer review history of their article (what does this mean?). If published, this will include your full peer review and any attached files.

**Do you want your identity to be public for this peer review?** For information about this choice, including consent withdrawal, please see our Privacy Policy.

Reviewer #1: No

---

## [Decision Letter · Decision Letter 1]

13 Mar 2024

PGPH-D-23-02074R1

Association between HTLV-1/2 infection and COVID-19 severity in a migrant Shipibo-Konibo population in Lima, Peru

Dear Dr. Avila,

Thank you for submitting your manuscript to PLOS Global Public Health. After careful consideration, we feel that it has merit but does not fully meet PLOS Global Public Health’s publication criteria as it currently stands. Therefore, we invite you to submit a revised version of the manuscript that addresses the points raised during the review process.

We look forward to receiving your revised manuscript.

Kind regards,

Charin Modchang, Ph.D.

Academic Editor

Journal Requirements:

Additional Editor Comments (if provided):

We have secured some additional reviewers and have now received their comments. Please address them accordingly.

Reviewers' comments:

Reviewer's Responses to Questions

**Comments to the Author**

1. If the authors have adequately addressed your comments raised in a previous round of review and you feel that this manuscript is now acceptable for publication, you may indicate that here to bypass the “Comments to the Author” section, enter your conflict of interest statement in the “Confidential to Editor” section, and submit your "Accept" recommendation.

Reviewer #2: (No Response)

Reviewer #3: All comments have been addressed

Reviewer #4: All comments have been addressed

2. Does this manuscript meet PLOS Global Public Health’s publication criteria? Is the manuscript technically sound, and do the data support the conclusions? The manuscript must describe methodologically and ethically rigorous research with conclusions that are appropriately drawn based on the data presented.

Reviewer #2: (No Response)

Reviewer #3: Yes

Reviewer #4: Yes

3. Has the statistical analysis been performed appropriately and rigorously?

Reviewer #2: (No Response)

Reviewer #3: Yes

Reviewer #4: Yes

4. Have the authors made all data underlying the findings in their manuscript fully available (please refer to the Data Availability Statement at the start of the manuscript PDF file)?

Reviewer #2: (No Response)

Reviewer #3: Yes

Reviewer #4: Yes

5. Is the manuscript presented in an intelligible fashion and written in standard English?

Reviewer #2: (No Response)

Reviewer #3: Yes

Reviewer #4: Yes

6. Review Comments to the Author

Reviewer #2: The study “Association between HTLV-1/2 infection and COVID-19 severity in a migrant Shipibo-Konibo population in Lima, Peru” is interesting and important. Fátima Avila Dextre and coauthors built a logistic regression model to estimate the risks of different factors. However, I have some concerns.

1. Line 261, The study has clear limitations as the authors already mentioned "...which adds uncertainty to whether the test was positive due to SARS-CoV-2 infection, false positivity, or vaccination". The study should control vaccination status, especially the authors cannot make sure whether positive test was due to natural infection or vaccination. Therefore, please also show the regression results after separating vaccinated and unvaccinated. Also show the proportion of participants among asymptomatic, mild and moderate for vaccinated and unvaccinated as Table 4.

2. Futhermore, line 113, "We used logistic regression to evaluate the association between HTLV-1/2 infection with severe COVID-19 and reported the odds ratio (OR)", authors should give more detailed descriptions of logistic model. Did authors performed variable selection or model comparison to determine an optimal set of variables?

3. The variable overcrowding in the model was not described and explained.

4. The difference of age distribution between HTLV(+) and HTLV(-) is limitation. Please discuss it.

Reviewer #3: In the revised manuscript "Association between HTLV-1/2 infection and COVID-19 severity in a migrant Shipibo-Konibo population in Lima, Peru" by Fátima Avila Dextre and colleagues, submitted to PLOS Global Public Health, the authors performed a study to describe the association between human T-cell lymphotropic virus type 1 and 2 (HTLV-1/2) infection, and the clinical severity of COVID-19, in the migrant community of the Shipibo-Konibo indigenous people in Lima, Peru. The infection with HTLV-1/2 is endemic in this population, which is a social vulnerable population, and fall within the scope of the journal. The authors did a cross-sectional descriptive study involving a survey of adult indigenous migrants residing in Cantagallo-Rímac, who were identified as suspected or confirmed cases of COVID-19, and found that in 182 individuals sampled had no significant association between HTLV-1/2 infection and the clinical severity of COVID-19. The questions from the previous reviewer were addressed, and I don´t have any concerns with this manuscript.

Reviewer #4: The authors have revised the manuscript and the important points are now clarified

7. PLOS authors have the option to publish the peer review history of their article (what does this mean?). If published, this will include your full peer review and any attached files.

**Do you want your identity to be public for this peer review?** For information about this choice, including consent withdrawal, please see our Privacy Policy.

Reviewer #2: No

Reviewer #3: No

Reviewer #4: No

---

## [Decision Letter · Decision Letter 2]

8 May 2024

PGPH-D-23-02074R2

Association between HTLV-1/2 infection and COVID-19 severity in a migrant Shipibo-Konibo population in Lima, Peru

Dear Dr. Avila,

Thank you for submitting your manuscript to PLOS Global Public Health. After careful consideration, we feel that it has merit but does not fully meet PLOS Global Public Health’s publication criteria as it currently stands. Therefore, we invite you to submit a revised version of the manuscript that addresses the points raised during the review process.

We look forward to receiving your revised manuscript.

Kind regards,

Charin Modchang, Ph.D.

Academic Editor

Journal Requirements:

Additional Editor Comments (if provided):

Thank you for submitting your revised manuscript. There is one remaining issue. The reviewer feels that the statistical part of the methods needed to be described clearly to ensure the study's reproducibility. Please address the comments accordingly.

Reviewers' comments:

Reviewer's Responses to Questions

**Comments to the Author**

1. If the authors have adequately addressed your comments raised in a previous round of review and you feel that this manuscript is now acceptable for publication, you may indicate that here to bypass the “Comments to the Author” section, enter your conflict of interest statement in the “Confidential to Editor” section, and submit your "Accept" recommendation.

Reviewer #2: (No Response)

2. Does this manuscript meet PLOS Global Public Health’s publication criteria? Is the manuscript technically sound, and do the data support the conclusions? The manuscript must describe methodologically and ethically rigorous research with conclusions that are appropriately drawn based on the data presented.

Reviewer #2: Partly

3. Has the statistical analysis been performed appropriately and rigorously?

Reviewer #2: I don't know

4. Have the authors made all data underlying the findings in their manuscript fully available (please refer to the Data Availability Statement at the start of the manuscript PDF file)?

Reviewer #2: No

5. Is the manuscript presented in an intelligible fashion and written in standard English?

Reviewer #2: Yes

6. Review Comments to the Author

Reviewer #2: Thanks for your revision. There are still few comments.

1. I cannot find your Supplementary material. Please make sure all your data or files are fully available.

2. In your response, "we have now explicitly described the methods used to determine the set of variables:

'Risk factors for the multivariable model were chosen based on clinical and

epidemiological relevance.' (Line 118-119)."

I appreciated that you have added more words to describe your model but this is still not clear enough. Please add your probability distribution, link function and regression model formula including all covariants. Did authors perform

variable selection or model comparison to determine an optimal set of variables? How to know model fit and how to decide which variables should be included or excluded? Please address these questions otherwise you may get different results.

7. PLOS authors have the option to publish the peer review history of their article (what does this mean?). If published, this will include your full peer review and any attached files.

**Do you want your identity to be public for this peer review?** For information about this choice, including consent withdrawal, please see our Privacy Policy.

Reviewer #2: No

---

## [Decision Letter · Decision Letter 3]

7 Jun 2024

PGPH-D-23-02074R3

Association between HTLV-1/2 infection and COVID-19 severity in a migrant Shipibo-Konibo population in Lima, Peru

Dear Dr. Avila,

Thank you for submitting your manuscript to PLOS Global Public Health. After careful consideration, we feel that it has merit but does not fully meet PLOS Global Public Health’s publication criteria as it currently stands. Therefore, we invite you to submit a revised version of the manuscript that addresses the points raised during the review process.

We look forward to receiving your revised manuscript.

Kind regards,

Charin Modchang, Ph.D.

Academic Editor

Journal Requirements:

Additional Editor Comments (if provided):

Thanks to the authors for addressing the reviewer's comments. The reviewer feels that the details of the statistical analysis still need to be revised for the research to be reproducible. Please be more specific when describing the analysis methodology.

Reviewers' comments:

Reviewer's Responses to Questions

**Comments to the Author**

1. If the authors have adequately addressed your comments raised in a previous round of review and you feel that this manuscript is now acceptable for publication, you may indicate that here to bypass the “Comments to the Author” section, enter your conflict of interest statement in the “Confidential to Editor” section, and submit your "Accept" recommendation.

Reviewer #2: All comments have been addressed

2. Does this manuscript meet PLOS Global Public Health’s publication criteria? Is the manuscript technically sound, and do the data support the conclusions? The manuscript must describe methodologically and ethically rigorous research with conclusions that are appropriately drawn based on the data presented.

Reviewer #2: Yes

3. Has the statistical analysis been performed appropriately and rigorously?

Reviewer #2: I don't know

4. Have the authors made all data underlying the findings in their manuscript fully available (please refer to the Data Availability Statement at the start of the manuscript PDF file)?

Reviewer #2: No

5. Is the manuscript presented in an intelligible fashion and written in standard English?

Reviewer #2: Yes

6. Review Comments to the Author

Reviewer #2: Thanks for the changes. I appreciate it. I have few more minor comments and I hope these can help to improve the manuscript.

May I suggest you to move your main description of the model in the supplementary information, starting from “We used logistic regression to evaluate the association between HTLV-1/2 infection with severe COVID-19, reporting the odds ratio (OR) with their corresponding 95% confidence intervals (CI)….” into the methods section in the main text. That will readers to understand your approach easily.

I still cannot find your source codes. Could you specify how to access your source codes.

About your variable selection, “Variable selection was guided by clinical and epidemiological relevance, supported by theoretical justifications and prior research findings (Garzón-Orjuela et al., 2022; Lima-Martínez et al., 2021; Liu et al., 2021; López-Tiro et al., 2022; Nakazaki et al., 2023; Pepera et al., 2022; Wang et al., 2022; Zheng et al., 2020)”, please be more specific to describe the approach. For example, what was the criteria and what variables were removed. I think this part should also be described in the main text to help readers to understand easily.

7. PLOS authors have the option to publish the peer review history of their article (what does this mean?). If published, this will include your full peer review and any attached files.

**Do you want your identity to be public for this peer review?** For information about this choice, including consent withdrawal, please see our Privacy Policy.

Reviewer #2: No

---

## [Editor Report · Decision Letter 4]

13 Jun 2024

Association between HTLV-1/2 infection and COVID-19 severity in a migrant Shipibo-Konibo population in Lima, Peru

PGPH-D-23-02074R4

Dear Ms. Avila,

We are pleased to inform you that your manuscript 'Association between HTLV-1/2 infection and COVID-19 severity in a migrant Shipibo-Konibo population in Lima, Peru' has been provisionally accepted for publication in PLOS Global Public Health.

Best regards,

Charin Modchang, Ph.D.

Academic Editor

The authors have thoroughly revised the manuscript and addressed the reviewers' comments through multiple rounds of revision. The manuscript now meets the standards for publication, and I recommend its acceptance in the current form. Thank you.